# Isometry pursuit

## Abstract

Isometry pursuit is a convex algorithm for identifying orthonormal column-submatrices of wide matrices. It consists of a vector normalization followed by multitask basis pursuit. Applied to Jacobians of putative coordinate functions, it helps identity locally isometric embeddings from within interpretable dictionaries. We provide theoretical and experimental results justifying this method, including a proof with realistic assumptions that such isometric submatrices, should they exist, are contained within the obtained support. For problems involving coordinate selection and diversification, it offers a synergistic alternative to greedy and brute force search.

## 1 Introduction

Many real-world problems may be abstracted as selecting a subset of the columns of a matrix representing stochastic observations or analytically exact data. This paper focuses on a simple such problem. Given a rank $D$ matrix $X \in \mathbb{R}^{D \times P}$ with $P > D$, select a square submatrix $X_{.S}$ where subset $S \subset [P]$ satisfies $|S| = D$ that is as orthonormal as possible.

This problem arises in interpretable learning specifically because while the coordinate functions of a given feature space may have no intrinsic meaning, it is sometimes possible to generate a dictionary of interpretable features which may be considered as potential parametrizing coordinates. When this is the case, selection of candidate interpretable features as coordinates can take the above form. While implementations vary across data and algorithmic domains, identification of such coordinates generally aids mechanistic understanding, generative control, and statistical efficiency.

This paper shows that an adapted version of the algorithm in Koelle et al. [2024] leads to a convex procedure that can improve upon greedy approaches such as those in Cai and Wang [2011], Chen and Meila [2019], Kohli et al. [2021], Jones et al. [2007] for finding isometries. The insight leading to isometry pursuit is that multitask basis pursuit applied to an appropriately normalized $X$ selects orthonormal submatrices. Given vectors in $\mathbb{R}^D$, the normalization log-symmetrizes length and favors those closer to unit length, while basis pursuit favors those which are orthogonal. Our results formalize this intuition within a limited setting, and show the usefulness of isometry pursuit as a trimming procedure prior to brute force search for diversification and interpretable coordinate selection. We also introduce a novel ground truth objective function against which we measure the success of our algorithm, and discuss the reasonableness of the trimming procedure.

## 2 Background

Our algorithm is motivated by spectral and convex analysis.

### 2.1 Problem

Our goal is, given a matrix $X \in \mathbb{R}^{D \times P}$, to select a subset $S \subset [P]$ with $|S| = D$ such that $X_{.S}$ is as orthonormal as possible in a computationally efficient way. To this end, we define a ground truth loss

---

[2]Code is available at `https://anonymous.4open.science/r/isometry-pursuit-8FC4/README.md`.

39th Conference on Neural Information Processing Systems (NeurIPS 2025).

function that measures orthonormalness, and then introduce a surrogate loss function that convexifies the problem so that it may be efficiently solved. Figure 1a and 1b provide a visual depiction of this problem when $D = 2$ and $P = 30$.

## 2.2 Interpretability and isometry

Our motivating example is the selection of data representations from within sets of putative coordinates: the columns of a provided wide matrix. Compared with Sparse PCA [d'Aspremont et al., 2007, Dey et al., 2017, Bertsimas and Kitane, 2022, Bertsimas et al., 2022], we seek a low-dimensional representation from the set of these column vectors rather than their span.

This method applies to interpretability, for which parsimony is at a premium. Interpretability arises through comparison of data with what is known to be important in the domain of the problem. This knowledge often takes the form of a functional dictionary. Evaluation of independence of dictionary features arises in numerous scenarios [Chen and Meilă, 2019, Koelle et al., 2022, He et al., 2023]. The requirement that dictionary features be full rank has been called functional independence [Koelle et al., 2022] or feature decomposability [Templeton et al., 2024], with connection between dictionary rank and independence via the implicit function theorem. Besides independence, the metric properties of such dictionary elements are of natural interest. This is formalized through the notion of differential.

**Definition 1** *The **differential** of a smooth map $\phi : \mathcal{M} \to \mathcal{N}$ between $D$ dimensional manifolds $\mathcal{M} \subseteq \mathbb{R}^B$ and $\mathcal{N} \subseteq \mathbb{R}^P$ is a map in tangent bases $x_1 \dots x_D$ of $T_\xi \mathcal{M}$ and $y_1 \dots y_D$ of $T_{\phi(\xi)} \mathcal{N}$ consisting of entries*

$$D\phi(\xi) = \begin{bmatrix} \frac{\partial \phi_1}{\partial x_1}(\xi) & \cdots & \frac{\partial \phi_1}{\partial x_D}(\xi) \\ \vdots & & \vdots \\ \frac{\partial \phi_D}{\partial x_1}(\xi) & \cdots & \frac{\partial \phi_D}{\partial x_D}(\xi) \end{bmatrix}. \tag{1}$$

It is not always necessary to explicitly estimate tangent spaces when applying this definition. The most commonly encountered manifolds are vector spaces for which the tangent spaces are trivial. This is the case for full-rank tabular data, for which isometry has a natural interpretation as a type of diversification, and often for the latent spaces of deep learning models. In this case, $B = D$.

**Definition 2** *A map $\phi$ between $D$ dimensional submanifolds with inherited Euclidean metric $\mathcal{M} \subseteq R^B$ and $\mathcal{N} \subseteq R^P$ $\phi$ is an **isometry at a point** $\xi \in \mathcal{M}$ if*

$$D\phi(\xi)^T D\phi(\xi) = I_D. \tag{2}$$

*That is, $\phi$ is an isometry at $\xi$ if $D\phi(\xi)$ is orthonormal.*

The applications of pointwise isometry are themselves manifold. Pointwise isometric embeddings faithfully preserve high-dimensional geometry. For example, Local Tangent Space Alignment [Zhang and Zha, 2004], Multidimensional Scaling [Chen and Buja, 2009] and Isomap [Tenenbaum et al., 2000] non-parametrically estimate embeddings that are as isometric as possible. Another approach stitches together pointwise isometries selected from a dictionary to form global embeddings [Kohli et al., 2021]. The method is particularly relevant since it constructs such isometries through greedy search, with putative dictionary features added one at a time.

That $D\phi$ is orthonormal has several equivalent formulations. The one motivating our ground truth loss function comes from spectral analysis.

**Proposition 1** *The singular values $\sigma_1 \dots \sigma_D$ are equal to 1 if and only if $U \in \mathbb{R}^{D \times D}$ is orthonormal.*

On the other hand, the formulation that motivates our convex approach is that orthonormal matrices consist of $D$ coordinate features whose gradients are orthogonal and of unit length.

**Proposition 2** *The component vectors $u_1 \dots u_D \in \mathbb{R}^B$ form a orthonormal matrix if and only if, for all $d_1, d_2 \in [D], \langle u_{d_1}, u_{d_2} \rangle = \begin{cases} 1 & d_1 = d_2 \\ 0 & d_1 \neq d_2 \end{cases}$.*

## 2.3  Best subset selection

Given a matrix $X \in \mathbb{R}^{D \times P}$, we compare algorithmic paradigms for solving problems of the form

$$\arg \min_{S \in \binom{[P]}{D}} l(X_{\cdot S}) \tag{3}$$

where $\binom{[P]}{D} = \{A \subseteq [P] : |A| = D\}$. Brute force algorithms consider all possible solutions. These algorithms are conceptually simple, but have the often prohibitive time complexity $O(C_l P^D)$ where $C_l$ is the cost of evaluating $l$. Greedy algorithms consist of iteratively adding one element at a time to $S$. This algorithms have time complexity $O(C_l PD)$ and so are computationally more efficient than brute force algorithms, but can get stuck in local minima. Formal definitions are given in Section 6.1.

Sometimes, it is possible to introduce an objective which convexifies problems of the above form [Liberti, 2004, Abdi, 2013, Zhou and Low, 2021]. Solutions

$$\arg \min f(\beta) : Y = X\beta \tag{4}$$

to the overcomplete regression problem $Y = X\beta$ are a classic example [Scott Shaobing Chen and David L. Donoho and Michael A. Saunders, 2001]. When $f(\beta) = \|\beta\|_0$, this problem is non-convex, and is thus suitable for greedy or brute algorithms, but when $f(\beta) = \|\beta\|_1$, the problem is convex, and may be solved efficiently via interior-point methods. When the equality constraint is relaxed, Lagrangian duality may be used to reformulate as a so-called Lasso problem, which leads to an even richer set of optimization algorithms [Hastie et al., 2016].

The form of basis pursuit that we apply is inspired by the group basis pursuit approach in Koelle et al. [2022]. In group basis pursuit (which we call multitask basis pursuit when grouping is dependent only on the structure of matrix-valued response variable $y$) the objective function is $f(\beta) = \|\beta\|_{1,2} := \sum_{p=1}^{P} \|\beta_{p\cdot}\|_2$ [Yuan and Lin, 2006, Obozinski et al., 2006, Yeung and Zhang, 2011]. This objective creates joint sparsity across entire rows $\beta_{p\cdot}$ and was used in Koelle et al. [2022] to select independent subsets among interpretable features.

# 3  Method

We apply the group lasso paradigm used to select independent dictionary elements in Koelle et al. [2022, 2024] to the more specific problem of selecting pointwise isometries from a dictionary. We first define a ground truth objective computable via brute and greedy algorithms that is uniquely minimized by orthonormal matrices. We then define the combination of normalization and multitask basis pursuit that approximates this ground truth loss function. We finally give a brute post-processing method for ensuring that the solution is $D$ sparse, and provide a theoretical result that the two stage approach will always result in recovery of an isometric solution from the dictionary, should one exist.

## 3.1  Ground truth

We'd like a ground truth objective to be minimized uniquely by orthonormal matrices, invariant under rotation, and depend on all changes in the matrix. Deformation [Kohli et al., 2021] and nuclear norm [Boyd and Vandenberghe, 2004] use only a subset of the differential's information and are not uniquely minimized at unitarity, respectively. We therefore introduce an alternative ground truth objective that satisfies the above desiderata and has convenient connections to isometry pursuit.

This ground truth objective is

$$l_c : \mathbb{R}^{D \times P} \to \mathbb{R}^+ \tag{5}$$

$$X \mapsto \sum_{d=1}^{D} g(\sigma_d(X), c) \tag{6}$$

where $\sigma_d(X)$ is the $d$-th singular value of $X$ and

$$g : \mathbb{R}^+ \times \mathbb{R}^+ \to \mathbb{R}^+ \tag{7}$$

$$t, c \mapsto \frac{e^{t^c} + e^{t^{-c}}}{2e}. \tag{8}$$

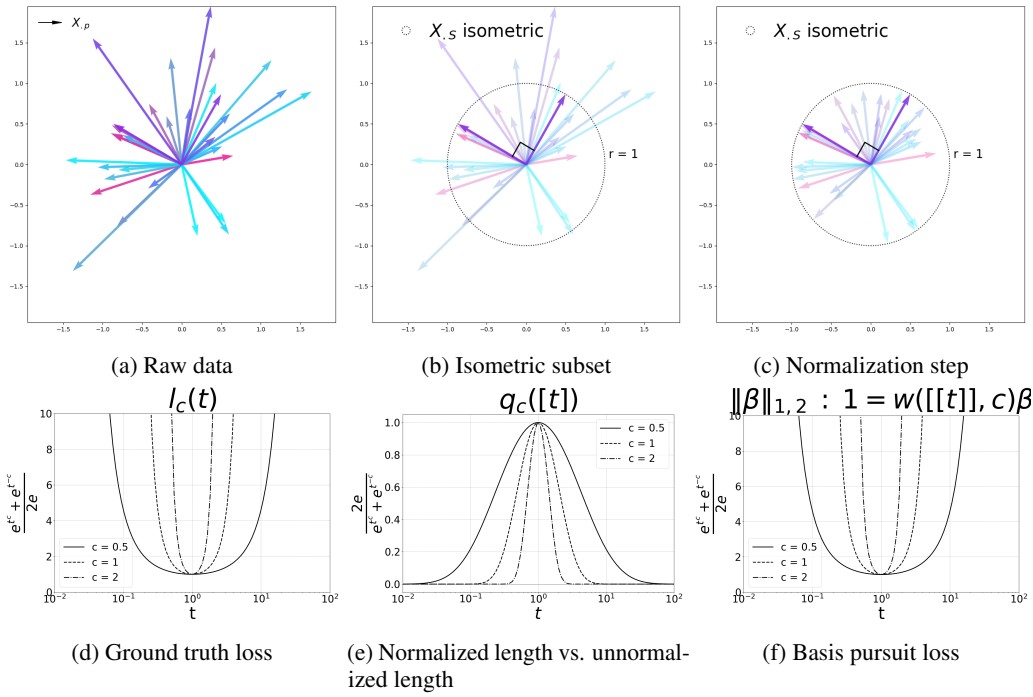

(a) Raw data       (b) Isometric subset       (c) Normalization step

(d) Ground truth loss    (e) Normalized length vs. unnormal-    (f) Basis pursuit loss
                       ized length

Figure 1: 1a Raw data $X$ represented as column-vectors. 1b An isometric subset - the identification of which is our objective. 1c The vectors after normalization so that vectors of length 1 maintain longest length. 1d Our ground truth loss function for different values of $c$ in the one-dimensional case $D = 1$ 1e Normalized length used to rescale Figure 1a to get vectors in Figure 1c. 1f Basis pursuit loss. This loss is equivalent to that shown in Figure 1d in the one-dimensional case.

By Proposition 1, we can see that $\ell_c$ is uniquely maximized by orthonormal matrices. Moreover, $g$ is convex, and $\ell_c(X^{-1}) = \ell_c(X)$ when $X$ is invertible. Figure 1d gives a graph of $l_c$ when $D = 1$.

Our ground truth objective is therefore the best subset selection problem

$$\arg \min_{S \in \binom{[P]}{d}} l_c(X_{\cdot S}). \tag{9}$$

Regardless of the convexity of $l_c$, brute combinatorial search over $[P]$ is inherently non-convex. This motivates our alternative formulation.

## 3.2 Normalization

Since basis pursuit methods tend to select longer vectors, selection of orthonormal submatrices requires normalization such that both long and short candidate basis vectors are penalized in the subsequent regression. We introduce the following definition.

**Definition 3 (Symmetric normalization)** *A function $q : \mathbb{R}^D \to \mathbb{R}^+$ is a symmetric normalization if*

$$\arg \max_{v \in \mathbb{R}^D} q(v) = \{v : \|v\|_2 = 1\} \tag{10}$$

$$q(v) = q\left(\frac{v}{\|v\|_2^2}\right) \tag{11}$$

$$q(v_1) = q(v_2) \ \forall \ v_1, v_2 \in \mathbb{R}^D : \|v_1\|_2 = \|v_2\|_2. \tag{12}$$

We use such functions to normalize column-vector length in such a way that vectors of length 1 prior to normalization have longest length after normalization and vectors are shrunk proportionately to

their deviation from 1. That is, we normalize vectors by

$$n : \mathbb{R}^D \to \mathbb{R}^D \tag{13}$$

$$v \mapsto q(v)v \tag{14}$$

and matrices by

$$w : \mathbb{R}^{D \times P} \to \mathbb{R}^D \tag{15}$$

$$X_{\cdot p} \mapsto n(X_{\cdot p}) \ \forall \ p \in [P]. \tag{16}$$

Given $c > 0$, we choose $q$ as follows.

$$q_c : \mathbb{R}^D \to \mathbb{R}^+ \tag{17}$$

$$v \mapsto \frac{e^{\|v\|_2^c} + e^{\|v\|_2^{-c}}}{2e}. \tag{18}$$

Besides satisfying the conditions in Definition 3, this normalization has some additional nice properties. First, $q$ is convex. Second, it grows asymptotically log-linearly. Third, while $\exp(-|\log t|) = \exp(-\max(t, 1/t))$ is a seemingly natural choice for normalization, it is non smooth, and the LogSumExp [Boyd and Vandenberghe, 2004] replacement of $\max(t, 1/t)$ with $\log(\exp(t) + \exp(1/t))$ simplifies to 18 upon exponentiation. Finally, the parameter $c$ grants control over the width of the basin, which may be useful for avoiding numerical issues arising close to 0 and $\infty$. We plot this normalization function in Figure 1e and its effect on our example data in Figure 1c.

### 3.3 Isometry pursuit

Isometry pursuit is the application of multitask basis pursuit to the normalized design matrix $w(X, c)$ to identify submatrices of $X$ that are as orthonormal as possible. Define the multitask basis pursuit penalty

$$\| \cdot \|_{1,2} : \mathbb{R}^{P \times D} \to \mathbb{R}^+ \tag{19}$$

$$\beta \mapsto \sum_{p=1}^{P} \|\beta_{p \cdot}\|_2. \tag{20}$$

Given a matrix $Y \in \mathbb{R}^{D \times D}$, the multitask basis pursuit solutions are

$$\widehat{\beta}_{MBP}(X, Y) := \arg \min_{\beta \in \mathbb{R}^{P \times D}} \|\beta\|_{1,2} \ : \ Y = X\beta. \tag{21}$$

Note that multitask basis pursuit solution is not unique under more relaxed conditions than for standard basis pursuit. Recall that, given a design matrix $X$, there exists a response variable $y$ such that the lasso admits a non-unique solution if and only if the rowspan of the design matrix $X$ contains a sufficient codimension edge of the $p$ dimensional hypercube [Schneider and Tardivel, 2022]. This condition generalizes the previously introduced general position condition - that affinely independent columns of the design matrix guarantee non-uniqueness. However, for Isometry Pursuit, non-unique solutions may occur even when this condition is satisfied. A simple example - $X = \begin{bmatrix} 1 & 0 & \frac{\sqrt{2}}{2} & \frac{\sqrt{2}}{2} \\ 0 & 1 & \frac{\sqrt{2}}{2} & \frac{-\sqrt{2}}{2} \end{bmatrix}$ results from the rotation invariance Proposition 4, but more subtle examples exist. We therefore define

$$\widehat{\beta}_{MLP}^{l_2}(X, Y) = \arg \min \|\beta\|_F \ : \ \beta \in \beta_{MBP}(X, Y). \tag{22}$$

By the strong convexity of the $l_2$ norm, $\widehat{\beta}_{MLP}^{l_2}(X, Y)$ is well defined.

This implicit regularization assumption is reasonable. For example, in Tibshirani [2012a] the implicit selection of the minimum $l_2$ norm solution among the lasso solutions was proven for the Least Angle Regression Selection (LARS) algorithm for solving the lasso, while Mishkin and Pilanci [2022a] assumes that the group lasso solution is in fact min-norm prior to subsequent theoretical analyses. Empirically, the Splitting Conic Solver type method (which is itself a variant of the

Alternating Direction Method of Multilpliers) [O'Donoghue et al., 2013] stably estimates the min-norm solution when initialized at 0, but besides a brief exploration in Section 6.4 we leave theoretical and experimental proof of this feature of the feature our optimization approach for future work, and instead make the ability of our optimizer to obtain the min-norm solution an assumption of our eventual proposition.

Isometry pursuit is then given by

$$\widehat{\beta}_c(X) := \widehat{\beta}_{MBP}^{l_2}(w(X,c), I_D) \tag{23}$$

where $I_D$ is the $D$ dimensional identity matrix and recovered functions are the indices of the dictionary elements with non-zero coefficients. That is, they are given by $S(\beta)$ where

$$S : \mathbb{R}^{P \times D} \to \binom{[P]}{D} \tag{24}$$

$$\beta \mapsto \{p \in [P] : \|\beta_{p.}\| > 0\}. \tag{25}$$

---

ISOMETRYPURSUIT(Matrix $X \in \mathbb{R}^{D \times P}$, scaling constant $c$)

1: Normalize $X_c = w(X, c)$
2: Optimize $\widehat{\beta} = \widehat{\beta}_{MBP}^{l_2}(X_c, I_D)$
3: **Output** $\widehat{S} = S(\widehat{\beta})$

---

### 3.4 Theory

The intuition behind our application of multitask basis pursuit is that submatrices consisting of vectors which are closer to 1 in length and more orthogonal will have smaller loss. A key theoretical assertion is that ISOMETRYPURSUIT is invariant to choice of basis for $X$.

**Proposition 3** *Let $U \in \mathbb{R}^{D \times D}$ be orthonormal. Then $S(\widehat{\beta}(UX)) = S(\widehat{\beta}(X))$.*

This proposition establishes the basis-independence of Isometry Pursuit and that we may replace $I_D$ in the constraint by any orthonormal $D \times D$ matrix. It is also essential for showing the following main theoretical result.

When a rank $D$ orthonormal column-submatrix $X_{.S}$ exists, the output of Program 23 will contain $S$.

**Proposition 4** *Let $X \in \mathbb{R}^{D \times P}$ have a rank $D$ orthonormal column submatrix $X_{.S}$. Then $S \subseteq \widehat{\beta}_c(X)$.*

Proofs of these propositions are given in Section 6.2, with corresponding experimental results in Section 4. The proof mostly relies on the KKT conditions [Hastie et al., 2015] being used to verify that an isometric subset yields a corresponding basis pursuit solution for an appropriate normalized matrix. This is a formalization of our intuition that longer and more orthogonal vectors will minimize loss. We then apply a simple argument to show that the $l_2$ minimizing solution is the least sparse solution, which means that it contains the isometry solution, should it exist.

### 3.5 Two-stage isometry pursuit

Proposition 4 suggests the following algorithm, which first uses the convex ISOMETRYPURSUIT algorithm to prune the $P$ candidate features and then applies brute search upon the reduced feature set. Similar two-stage approaches are standard in the Lasso literature Hesterberg et al. [2008], Koelle et al. [2022], and Proposition 4 puts them on relatively strong footing in our application, since this approach is guaranteed to select an isometric submatrix should one exist. This method forms our practical isometry estimator, the core advantage of which over purely brute search is a substantially reduced feature set.

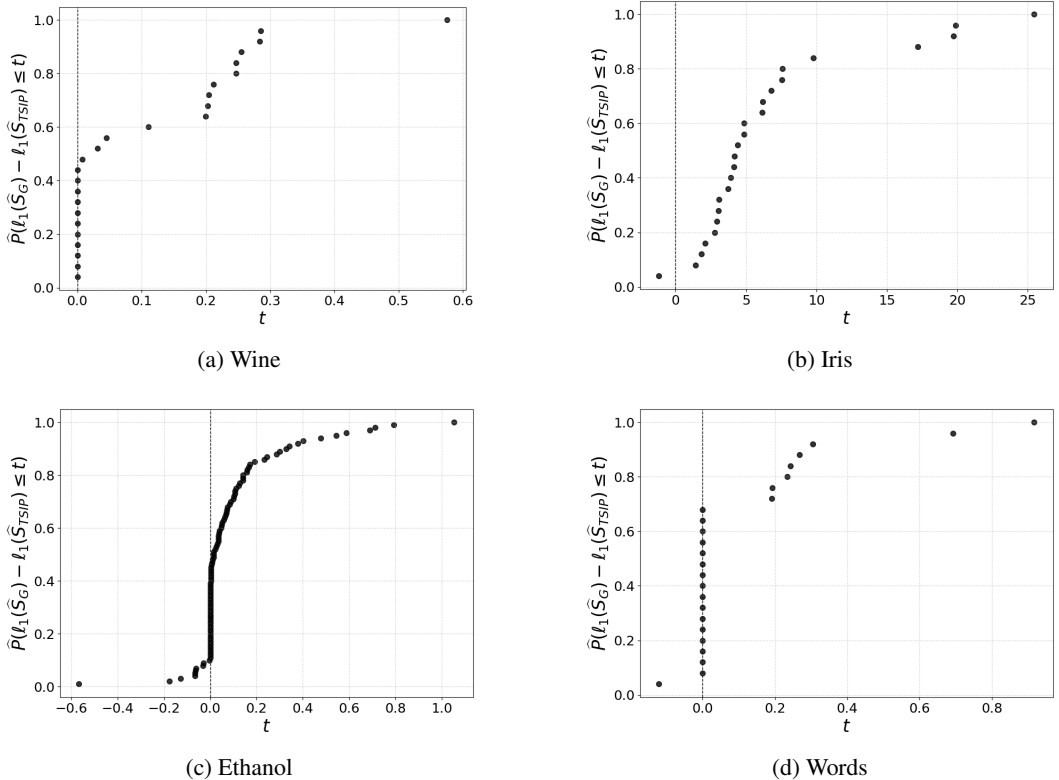

| (a) Wine | (b) Iris |
|---|---|
| (c) Ethanol | (d) Words |

Figure 2: Empirical cumulative distributions of isometry losses $\ell_1(\widehat{S}_g) - \ell_1(\widehat{S}_{TSIP})$ for Wine, Iris, Ethanol, and Words datasets across $R$ replicates. The distributions reflect the relative $\ell_1$ losses of the supports estimated by greedy optimization on the $\ell_1$ objective is less than or equal to $\ell_1$ loss of the support estimated by TWOSTAGEISOMETRYPURSUIT. As shown here and detailed in Table 1, losses are generally lower for two-stage isometry pursuit solutions.

---

TWOSTAGEISOMETRYPURSUIT(Matrix $X \in \mathbb{R}^{D \times P}$, scaling constant $c$)

1: $\widehat{S}_{IP} = $ ISOMETRYPURSUIT$(X, c)$
2: $\widehat{S} = $ BRUTESEARCH$(X_{\cdot \widehat{S}_{IP}}, l_c)$
3: **Output** $\widehat{S}$

---

## 4 Experiments

Say you are hosting an elegant dinner party, and wish to select a balanced set of wines for drinking and flowers for decoration. We demonstrate TWOSTAGEISOMETRYPURSUIT and GREEDYSEARCH on the Iris and Wine datasets [Fisher, 1988, Aeberhard and Forina, 1991, Pedregosa et al., 2011]. This has an intuitive interpretation as selecting diverse elements that reflects the peculiar structure of the diversification problem. Features like *petal width* are rows in $X$. They are features on the basis of which we may select among the flowers those which are most distinct from another. Thus, in diversification, $P = n$.

Our main interpretability dataset is the Ethanol dataset from Chmiela et al. [2018], Koelle et al. [2022]. Tather than selecting between bourbon and scotch we evaluate a dictionary of interpretable features - bond torsions - for their ability to parameterize the molecular configuration space. In this interpretability use case, columns denote gradients of informative features. We compute Jacoban matrices of putative parametrization functions and project them onto estimated tangent spaces (see Koelle et al. [2022] for preprocessing details). Rather than selecting between data points, we are selecting between functions which parameterize the data. This dataset exemplifies our motivating example - the search for locally isometric embeddings.

| Name | $D$ | $P$ | $R$ | $c$ | $\ell_1(X_{.\widehat{S}_G})$ | $|\widehat{S}|$ | $\ell_1(X_{.\widehat{S}})$ | $\widehat{P}(\ell_1(X_{.\widehat{S}_G}) > \ell_1(X_{.\widehat{S}}))$ | $\widehat{P}(\ell_1(X_{.\widehat{S}_G}) = \ell_1(X_{.\widehat{S}}))$ | $\widehat{P}(\bar{\ell}_1(X_{.\widehat{S}_G}) > \bar{\ell}_1(X_{.\widehat{S}}))$ |
|---|---|---|---|---|---|---|---|---|---|---|
| Iris | 4 | 75 | 25 | 1 | $14 \pm 7.3$ | $6.7 \pm 1$ | $6.9 \pm 1.4$ | 0.96 | 0 | 2.4e-05 |
| Wine | 6 | 89 | 25 | 1 | $7.7 \pm 0.33$ | $13 \pm 1.5$ | $7.6 \pm 0.29$ | 0.64 | 0.16 | 6.3e-04 |
| Ethanol | 2 | 756 | 100 | 1 | $2.6 \pm 0.3$ | $90 \pm 1.7e{+}02$ | $2.5 \pm 0.2$ | 0.66 | 0.17 | 2.1e-05 |
| Words | 6 | 61 | 25 | 1 | $14 \pm 1.3$ | $11 \pm 1.3$ | $14 \pm 1.2$ | 0.52 | 0.12 | 2.1e-02 |

Table 1: Experimental parameters and results. For Iris, Wine, and Words, probability estimates $\widehat{P}$ result from random downsampling features $[P]$ by a factor of 2 to create $R$ replicates, while for Ethanol, replicates correspond to individual data points. Distributional probabilities $\widehat{P}(\ell_1(X_{.\widehat{S}_G}) > \ell_1(X_{.\widehat{S}_{TSIP}}))$ and $\widehat{P}(\ell_1(X_{.\widehat{S}_G}) = \ell_1(X_{.\widehat{S}_{TSIP}}))$ are empirical across replicates, while asymptotic probabilities $\widehat{P}(\bar{\ell}_1(X_{.\widehat{S}_G}) > \bar{\ell}_1(X_{.\widehat{S}_{TSIP}}))$ is computed by paired two-sample T-test on $\ell_1(X_{.\widehat{S}})$ and $\ell_1(X_{.\widehat{S}_G})$. For brevity, in this table $\widehat{S} := \widehat{S}_{TSIP}$.

We also construct a small word embedding dataset. Inspired by the linear representation hypothesis [Park et al., 2023] and the construction over complete dictionaries of concepts in representation space [Templeton et al., 2024, Makelov et al., 2024], we apply isometric pursuit to prune down embeddings of concept dictionaries into their basis components. As with the other datasets, we measure success primarily numerically against the ground truth objective values obtained by greedy solutions. Further details on this dataset are given in Section 6.5.

For basis pursuit, we use the SCS interior point solver [O'Donoghue et al., 2016] from CVXPY [Diamond and Boyd, 2016, Agrawal et al., 2018], which is able to push sparse values arbitrarily close to 0 [CVXPY Developers]. Statistical replicas for Wine, Iris, and Words are created by resampling across $[P]$. Due to differences in scales between rows, these are first standardized. For the Wine dataset, even BRUTESEARCH on $\widehat{S}_{IP}$ is prohibitive in $D = 13$, and so we truncate our inputs to $D = 6$. For Ethanol, replicas are created by sampling from data points and their corresponding tangent spaces are estimated in $B = 252$.

Figure 2 and Table 1 show that the $\ell_1$ accrued by the subset $\widehat{S}_G$ estimated using GREEDYSEARCH with objective $\ell_1$ is higher than that for the subset estimated by TWOSTAGEISOMETRYPURSUIT. This effect is statistically significant across datasets. Generally, the ground truth loss obtained by GREEDYSEARCH optimization of $\ell_1$ is higher and therefore worse than that obtained by two-stage isometry pursuit. Table 1 and Figure 4 detail intermediate support recovery cardinalities from ISOMETRYPURSUIT. Isometry pursuit substantially reduced the size of the dictionary and made brute search computationally feasible. Wall-clock runtimes are given in Section 6.6. However, the preferable performance of two-stage isometry pursuit is not robust to different choices of $c$ on the Wine dataset in Figure 3. These show that the preference for the two-stage isometry pursuit solution is strongest around $c = 1$ and is consistent across feature truncation dimensions.

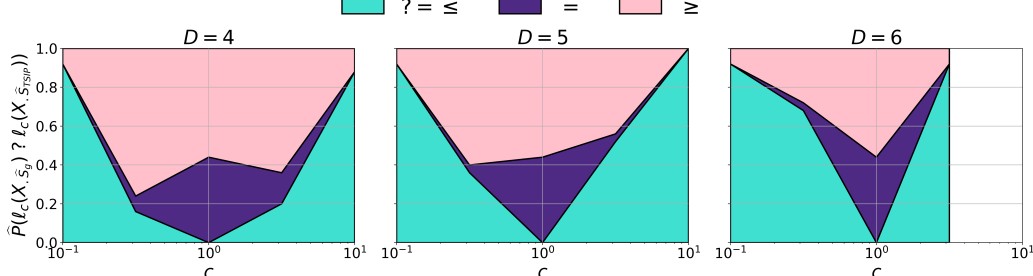

Figure 3: Proportions of support selection outcomes as power parameter $c$ varies, for different values of $D$ in the Wine dataset across various values of $c$. The turquoise area indicates when greedy solution $\widehat{S}_G$ outperforms $\widehat{S}_{\text{TSIP}}$, purple shows ties, and pink indicates $\widehat{S}_{\text{TSIP}}$ is better. Solution at $c = 10$ is not plotted for $D = 6$ due to numerical overflow.

## 5   Discussion

We have shown that multitask basis pursuit can help select isometric submatrices from appropriately normalized wide matrices. This approach - isometry pursuit - is a convex alternative to greedy methods for selection of orthonormalized features from within a dictionary. Isometry pursuit can be applied to diversification and geometrically-faithful coordinate estimation. Our experiments exemplify these applications, but more can be done.

One potential application is diversification in recommendation systems [Carbonell and Goldstein, 1998a, Wu et al., 2019, Lan] and other retrieval systems such as in RAG [Gao et al., 2023, Pickett et al., 2024, In et al., 2024, Weiss, 2024, Vec]. This is particularly relevant as maximum cosine similarity used in retrieval corresponds to the first coefficient of the Lasso regularization path [Koelle et al., 2022]. Compared with the greedy algorithms used in diversification [Carbonell and Goldstein, 1998b, Barioni et al., 2011, Drosou and Pitoura, 2010, Qin et al., 2012, Kunaver and Požrl, 2017, Guo and Sanner, 2010, Abdool et al., 2020, Yu et al., 2016, Huang et al., 2024, Pickett et al., 2024, Cai and Wang, 2011, Zhu et al., 2020], the convex reformulation may add speed and convergence to a global minima. However, we note that the standard max-sum diversity formulation that seeks to identify datapoints that are as far apart as possible is substantially different from the notion of diversity selected for by isometry pursuit [Kuo et al., 1993, Ghosh, 1996, Alfonso Cevallos, Friedrich Eisenbrand, and Rico Zenklusen, 2016, Ashkan et al., 2015].

The comparison of greedy [Mallat and Zhang, 1993, 1992, Pati et al., 1993, Tropp et al., 2005] and convex [Scott Shaobing Chen and David L. Donoho and Michael A. Saunders, 2001, Tropp, 2006, Chen and Huo, 2006] basis pursuit formulations has a rich history, and theoretical understanding of the behavior of this approximation is evolving. For example, that the solution of a lasso problem can sometimes be a non-singleton set is well-known [Osborne et al., 2000, Dossal, 2012, Chrétien and Darses, 2011, Tibshirani, 2012b, Ewald and Schneider, 2017, Ali and Tibshirani, 2018, Schneider and Tardivel, 2020, Mishkin and Pilanci, 2022b, Dupuis and Vaiter, 2019, Debarre et al., 2020, Everink et al., 2024], but Isometry Pursuit can generate non-unique solutions even when the design matrix satisfies general position. The nature of the approximation to the $\ell_0$ solution for isometry pursuit is also different than for standard Lasso theory. The main theoretical question of isometry pursuit is how well the minimizer of a convex loss approximates the singular value loss, rather than how well the convex loss performs in statistical estimation.

Algorithmic variants of interest include the multitask lasso [Hastie et al., 2015] extension of our estimator, as well as characterization of $D$ function selection within $\mathbb{R}^B$. Tangent-space specific variants have been studied in more detail in Koelle et al. [2022, 2024] with additional grouping across datapoints, and a corresponding variant of the isometry theorem that missed non-uniqueness was claimed in Koelle [2022]. Comparison of our loss with curvature - whose presence prohibits $D$ element isometry - could also prove fertile, as could application of convex matrix inversion via multitask basis pursuit.

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

# 6 Supplement

This section contains algorithms, proofs, and experiments in support of the main text.

## 6.1 Algorithms

We give definitions of the brute and greedy algorithms for the combinatorial problem studied in this paper. The brute force algorithm is computationally intractable for all but the smallest problems, but always finds the global minima.

---

BRUTESEARCH(Matrix $X \in \mathbb{R}^{D \times P}$, objective $f$)

---

1: **for** each combination $S \subseteq \{1, 2, \ldots, P\}$ with $|S| = D$ **do**
2:     Evaluate $f(X_{.S})$
3: **end for**
4: **Output** the combination $S^*$ that minimizes $f(X_{.S})$

---

Greedy algorithms are computationally expedient but can get stuck in local optima [Cormen et al., 2009, Russell and Norvig, 2009], even with randomized restarts [Dick et al., 2014].

---

GREEDYSEARCH(Matrix $X \in \mathbb{R}^{D \times P}$, objective $f$, selected set $S = \emptyset$, current size $d = 0$)

---

1: **if** $d = D$ **then**
2:     **Return** $S$
3: **else**
4:     **Initialize** $S_{\text{best}} = S$
5:     **Initialize** $f_{\text{best}} = \infty$
6:     **for** each $p \in \{1, 2, \ldots, P\} \setminus S$ **do**
7:         **Evaluate** $f(X_{.(S \cup \{p\})})$
8:         **if** $f(X_{.(S \cup \{p\})}) < f_{\text{best}}$ **then**
9:             **Update** $S_{\text{best}} = S \cup \{p\}$
10:            **Update** $f_{\text{best}} = f(\mathcal{X}_{.(S \cup \{p\})})$
11:         **end if**
12:     **end for**
13:     **Return** GREEDYSEARCH$(X, f, S_{\text{best}}, d + 1)$
14: **end if**

---

## 6.2 Proofs

### 6.2.1 Proof of Proposition 3

In this proof we first show that the penalty $\|\beta\|_{1,2}$ is unchanged by unitary transformation of $\beta$.

**Proposition 5** *Let $U \in \mathbb{R}^{D \times D}$ be unitary. Then $\|\beta\|_{1,2} = \|\beta U\|$.*

**Proof:**

$$\|\beta U\|_{1,2} = \sum_{p=1}^{P} \|\beta_{p.} U\| \tag{26}$$

$$= \sum_{p=1}^{P} \|\beta_{p.}\| \tag{27}$$

$$= \|\beta\|_{1,2} \tag{28}$$

$$\square$$

We then show that this implies that the resultant loss is unchanged by unitary transformation of $X$.

**Proposition 6** *Let $U \in \mathbb{R}^{D \times D}$ be unitary. Then $\widehat{\beta}(UX) = \widehat{\beta}(X)U$.*

**Proof:**

$$\widehat{\beta}(UX) = \arg\min_{\beta \in \mathbb{R}^{P \times D}} \|\beta\|_{1,2} \;:\; I_D = UX\beta \tag{29}$$

$$= \arg\min_{\beta \in \mathbb{R}^{P \times D}} \|\beta\|_{1,2} \;:\; U^{-1}U = U^{-1}UX\beta U \tag{30}$$

$$= \arg\min_{\beta \in \mathbb{R}^{P \times D}} \|\beta\|_{1,2} \;:\; I_D = X\beta U \tag{31}$$

$$= \arg\min_{\beta \in \mathbb{R}^{P \times D}} \|\beta U\|_{1,2} \;:\; I_D = X\beta U \tag{32}$$

$$= \arg\min_{\beta \in \mathbb{R}^{P \times D}} \|\beta\|_{1,2} \;:\; I_D = X\beta. \tag{33}$$

$\square$

### 6.2.2 Proof of Proposition 4

In this proof we first show that the orthonormal submatrix $X_{.S}$ is contained within the set $\widehat{\beta}_{MBP}(w(X, c), I_D)$.

**Proposition 7** *Let $X_{.S}$ be a rank $D$ orthonormal submatrix of $X$. Then $S \in S(\beta) \;:\; \beta \in \widehat{\beta}_{MBP}(w(X, c), I_D))$.*

**Proof:** By Proposition 3 without loss of generality, let $X_{.S} = I_D$. Also, without loss of generality let $S = \{1 \ldots D\}$. We show that $\beta = [I_D \; 0]$ satisfies the KKT conditions for $X = [I_D \; X_{-S}]$.

Recall that the KKT conditions for a convex optimization problem are a set of conditions on an element of the domain of the optimization algorithm that, if satisfied, certify that the element is a solution. To write out the KKT conditions, we define the Lagrangian

$$\mathcal{L}(X, \beta, \nu) = \|\beta\|_{1,2} + \nu^T(I_D - w(X, c)\beta) \tag{34}$$

where $\nu \in \mathbb{R}^{D \times D}$ is the dual variable.

The KKT conditions are then

- Primal feasibility: $w(X, c)\beta = I_D$
- Stationarity: there exists a dual variable $\nu$ such that $0 \in \partial\|\beta\|_{1,2} - w(X, c)^T\nu$ where $\partial$ is the subdifferential operator.

where dual feasibility and complementary slackness are ignorable by virtue of the absence of inequality constraints.

First note that in our case, primal feasibility is satisfied by $X$ being rank $D$ and $w$ not effecting the rank of the transformed matrix since it only rescales length and not direction.

For stationarity, recall that

$$\|\beta\|_{1,2} = \sum_{p=1}^{P} \|\beta_{p.}\|_2. \tag{35}$$

Then, by the definition of subdifferential

$$\partial\|\beta\|_{1,2} = \begin{bmatrix} I_D v_{d+1} \\ \ldots \\ v_P \end{bmatrix} \tag{36}$$

where $v_p \in \mathbb{R}^D$ can be any vector satisfying $\|v_p\|_2 \leq 1$.

We therefore must show that there exists a $\nu$ that satisfies

$$\begin{bmatrix} I_D \\ w(X_{-S}, c)^T \end{bmatrix} \nu = \begin{bmatrix} I_D \\ V_{-S} \end{bmatrix}. \tag{37}$$

At this point we can see that in fact $I_D$ is an appropriate choice of $\nu$ since normalization by $w$ leads to vectors satisfying $\|w(X, c\|_2 \leq 1$.

Since for this $\beta$, $S(\beta) = S$, we have shown that $S$ is the support of a solution to $\beta_{MBP}(w(X,c), I_D)$.
$\square$

The next part of the proof handles the presence of non-unique solutions.

**Proposition 8** $S(\widehat{\beta}_c(X)) = \cup \, S(\beta) \; : \; \beta \in \beta_{MBP}(w(X,c), I_D)$

**Proof:** The proposition follows from geometric properties of the solution set. We ignore the case for singleton solutions, for which the proposition is self-evident. Recall that $\widehat{\beta}_c(X) :=$ $\widehat{\beta}_{MBP}^{l_2}(w(X,c), I_D) = \arg\min_{\beta \in \mathbb{R}^{P \times D}} \|\beta\|_F \; : \; \beta \in \widehat{\beta}_{MBP}(X, I_D)$. By definition of $\widehat{\beta}_{MBP}(X, I_D)$, $\|\beta_{p.}\|_2 : \beta \in \widehat{\beta}_{MBP}(X, I_D)$ is an affine linear subspace of $\mathbb{R}^P$. The projection of the vector of minimum $l_2$ distance from the origin to a affine space intersects that space perpendicularly. However, for all $\|\beta_{.p}\|_2$ such that $\beta$ is on the intersection of this affine space with the coordinate hyperplanes formed by fixing any choices of $\|\beta_{.p}\|_2 = 0$, this vector doesn't intersect this space perpendicularly. Thus, $\widehat{\beta}_c(X)$ is not on the intersection of the solution polytope $\widehat{\beta}_{MBP}(w(X,c), I_D)$ with the coordinate hyperplanes. It is therefore maximally non-sparse with respect to any of the minimizing solutions. $\square$

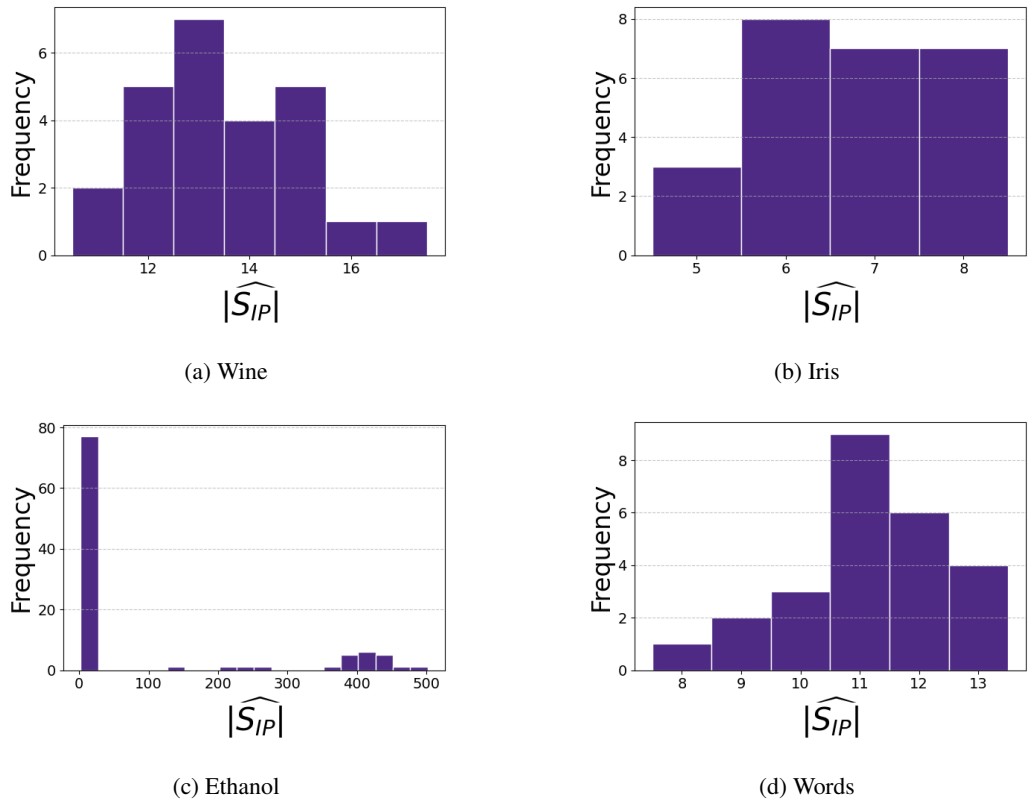

(a) Wine

(b) Iris

(c) Ethanol

(d) Words

Figure 4: Support cardinalities for each dataset after ISOMETRYPURSUIT.

## 6.3 Support cardinalities

Figure 4 plots the distribution of $|\widehat{S}_{IP}|$ from Table 1 in order to contextualize the reported means. While typically $|\widehat{S}_{IP}| << P$, there are cases for Ethanol where this is not the case that drive up the means. Further investigation into the extreme failure of isometry pursuit to sparsify in the the minor chunk of the bimodal Ethanol support cardinality distribution is warranted.

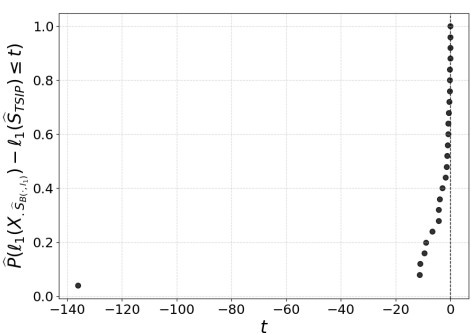

(a) Iris $\ell_1(X_{.\widehat{S}})$ comparison for BRUTE-SEARCH optimization of $\ell_1$ versus TWOSTAGEISOMETRYPURSUIT optimization of $\ell_1$

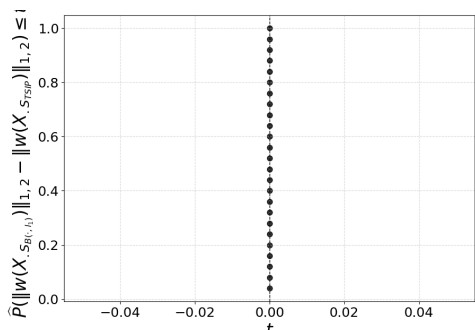

(b) Iris $\|X_{.\widehat{S}}\|_{1,2}$ comparison for BRUTE-SEARCH optimization of $\|\|\|_{1,2}$ versus TWOSTAGEISOMETRYPURSUIT optimization of $\|\|\|_{1,2}$

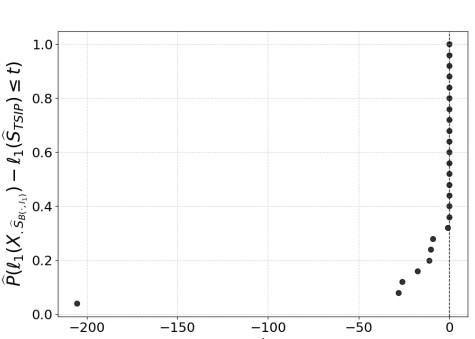

(c) Wine $\ell_1(X_{.\widehat{S}})$ comparison for BRUTE-SEARCH optimization of $\ell_1$ versus TWOSTAGEISOMETRYPURSUIT optimization of $\ell_1$

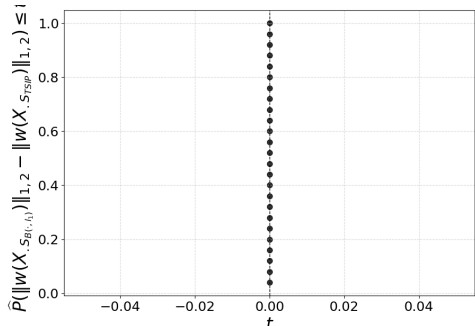

(d) Wine $\|X_{.\widehat{S}}\|_{1,2}$ comparison for BRUTE-SEARCHoptimization of $\|\|\|_{1,2}$ versus TWOSTAGEISOMETRYPURSUIT optimization of $\|\|\|_{1,2}$

Figure 5: Comparison of Isometry and Group Lasso Losses across 25 replicates for randomly downsampled Iris and Wine Datasets with $(P, D) = (4, 15)$ and $(13, 18)$, respectively. Note that this further downsampling compared with Section 4 was necessary to compute global minimizers of BRUTESEARCH.

## 6.4 Proposition 4 deep dive

The results in Section 4 show that there are circumstances in which the GREEDYSEARCH performs better than TWOSTAGEISOMETRYPURSUIT, so clearly TWOSTAGEISOMETRYPURSUIT does not always achieve a global optimum. Figure 5 gives results on the line of inquiry about why this is the case based on the reasoning presented in Section 5. In these results a two-stage algorithm achieves the global optimum of a slightly different brute problem, namely brute optimization of the multitask basis pursuit penalty $\|\cdot\|_{1,2}$. That is, brute search on $\|\cdot\|_{1,2}$ gives the same result as the two stage algorithm with brute search on $\|\cdot\|_{1,2}$ subsequent to isometry pursuit. This suggests that failure to select the global optimum by TWOSTAGEISOMETRYPURSUIT is due to the mismatch between global optimums of brute optimization of the multitask penalty and the isometry loss given certain data. It also confirms that the minimizer of brute optimization of the $\|\|\|_{1,2}$ objective is contained within the isometry pursuit solution. Note that by Proposition 4, this would include an isometric solution, should it exist.

| Pole 1 | Pole 2 | Category | Words (Pole 1) | Words (Pole 2) |
|--------|--------|----------|----------------|----------------|
| man | woman | gender | man, male, guy, gentleman, sir | woman, female, lady, madam, gal |
| regent | slave | status | regent, sovereign, monarch, emperor, pharaoh | slave, wench |
| young | old | age | young, kid | old, elderly |
| warrior | scholar | domain | warrior, soldier, knight, fighter, mercenary | scholar, academic, professor, intellectual, scientist |
| rich | poor | wealth | rich, wealthy, affluent, millionaire, tycoon | poor, impoverished, broke, destitute, beggar |
| urban | rural | location | urban, city, metropolitan, cosmopolitan, downtown | rural, village, countryside, agrarian, pastoral |
| warm | cold | warmth | warm, friendly, kind, loving, affectionate | cold, aloof, harsh, distant, indifferent |

Table 2: Semantic axes used for projection of word embeddings.

## 6.5 Word embeddings

We applied ISOMETRYPURSUIT on a word embedding dataset with the following construction. We used all-MiniLM-L6-v2 from the Sentence Transformers library [Reimers and Gurevych, 2019]. We defined 6 axes of variation using the word categories shown in Table 2. Each axis is defined by the difference between two poles, computed from the centroids of the embeddings of the corresponding word sets. We then projected the word embeddings onto these axes and standardized in order to create a linear conceptual embedding space. This was the representation that we evaluated.

In order to illustrate the semantic interpretation of our analyses, we plot the following co-occurrence graph for retrieved words. While we do not provide any quantitative evaluation beyond those in Section 4, in the plotted replicate two-stage isometry pursuit better separates words from the same category (e.g. cold, warm) than does greedy search.

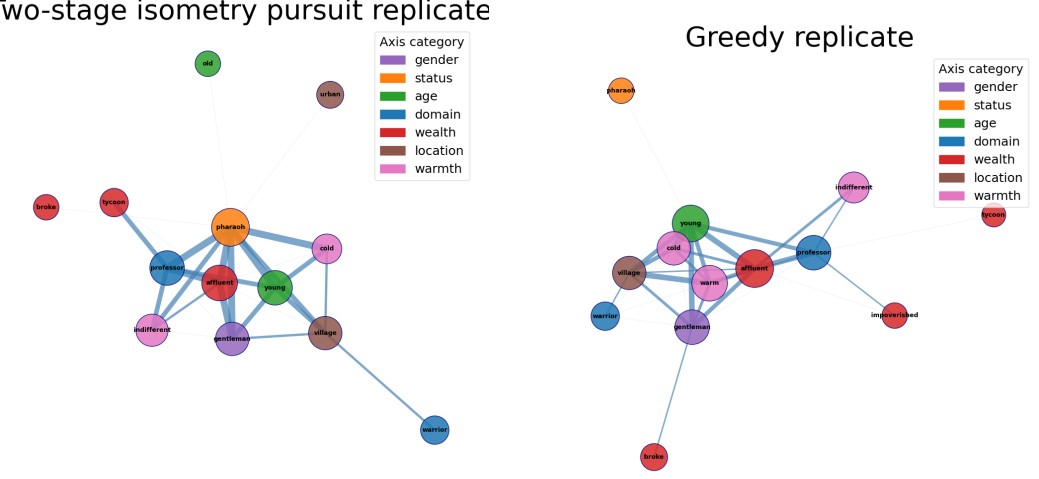

(a) Two-stage isometry pursuit selected word co-occurrences

(b) Greedy algorithm selected word co-occurrences

Figure 6: Force-directed graphs of word co-occurrences for each selection method. Node size reflects marginal frequency across replicates. Edge width represents co-occurrence strength ($\geq 5$). Colors denote semantic axis.

## 6.6 Timing

While wall-time of algorithms is a non-theoretical quantity that depends on implementation details, it provides valuable context for practitioners. We therefore report the following runtimes on a 2021 Macbook Pro. The particularly high variance for brute force search in the second step of TWOSTAGEISOMETRYPURSUIT is likely due to the large cardinalities reported in Figure 4.

| Name | IP | 2nd stage | Greedy |
|---|---|---|---|
| Iris | 1.24 ± 0.02 | 0.00 ± 0.00 | 0.02 ± 0.00 |
| Wine | 2.32 ± 0.17 | 0.13 ± 0.12 | 0.03 ± 0.00 |
| Ethanol | 8.38 ± 0.57 | 0.55 ± 1.08 | 0.07 ± 0.01 |
| Words | 0.98 ± 0.20 | 0.03 ± 0.02 | 0.01 ± 0.00 |

Table 3: Algorithm runtimes in seconds across replicates.

