# OpenReview forum: "Isometry pursuit"
_NeurIPS.cc/2025/Conference — Submitted to NeurIPS 2025_

### Official Review · Reviewer_h74u · 2025-07-01

**Clarity:** 3
**Significance:** 2
**Originality:** 2
**Rating:** 3
**Confidence:** 4

**Summary:**

The paper proposes Isometry Pursuit which is a convex approach to identify near-orthonormal subsets of columns from a wide matrix using normalization and multitask basis pursuit (MBP). The key theoretical contribution is a guarantee that if a true isometric (orthonormal) submatrix exists, then it is contained in the support of the relaxed MBP solution. The method is compared to a greedy baseline and evaluated on some datasets (such as Wine and Iris).

**Questions:**

1) Motivation: Why is orthonormality the right or sufficient condition for interpretability or coordinate selection? How does this help in practice for downstream tasks (e.g., learning, generalization)?
2) Theory: Is the solution approximately orthogonal in the approximate case where no isometry exists? Is there a bound on the deviation from unitarity? Can you derive such results?
3) Experiments: Comparisons to other subset selection methods

**Ethical Concerns:**

["NO or VERY MINOR ethics concerns only"]

**Final Justification:**

Based on the authors's response and the discussions, my concerns remain, especially regarding the limited novelty, the lacking motivation and the major limitation concerning orthogonality. Given these issues, I stick to my rating. The work needs more in my opinion to meet the bar of acceptance.

**Paper Formatting Concerns:**

No concerns

**Quality:**

2

**Strengths And Weaknesses:**

Strengths

+ The paper has a clear theoretical goal. In particular, it defines a clean and interpretable optimization problem: select D columns from a P-column matrix to approximate an orthonormal basis.
+ The relaxation via multitask basis pursuit is standard but applied here in a targeted way. The main theorem (Proposition 4) states that if an orthonormal submatrix exists, it is in the support of the MBP solution. This is a good result under the assumption that such a submatrix exists.
+ The experimental comparisons with greedy show modest gains.

Weaknesses
- Limited novelty: The method largely builds on known tools. In particular, the formulation is a direct application of group Lasso/multitask BP to a normalized design matrix. Also, the main theoretical guarantee is similar in spirit to known results in sparse recovery, especially in group sparsity contexts. The idea of trimming using convex relaxation before brute-force or greedy is standard in Lasso literature and feature selection. The paper feels more of a repackaging of known techniques and not a fundamentally new algorithmic or theoretical insight.
- I found the motivation to be underdeveloped. The ties of the method proposed to interpretability, diversification, and embedding tasks are loose and not rigorously explored. For example, why is orthonormality the right or sufficient condition for interpretability or coordinate selection? How does this help in practice for downstream tasks (e.g., learning, generalization)? The introduction and discussion handwave at these points but do not really deliver on them, which weakens the practical relevance of the contribution.
- The theory only applies when a true isometry exists. but no bounds are given on how well the method performs when no exact orthonormal subset exists. This is a major limitation. For instance, is the solution approximately orthogonal? Is there a bound on the deviation from orthogonality? Without such results, the utility of the algorithm in the general case is unclear.
- The experiments are too simplistic. The datasets (Wine, Iris, Ethanol) are small and toy-like. There is no compelling large-scale use case shown. Also, comparisons are limited to simple greedy, but no comparisons are made to more modern subset selection methods. The runtime analysis is limited and does not rigorously show why the proposed method is superior.
- There are several points where definitions or symbols are introduced abruptly or used vaguely. For example, the differential definitions (Def. 1 and 2) are not tightly integrated into the rest of the math. The normalization function q is defined in a confusing way.
- Even if correct and moderately effective, the method does not offer a breakthrough for sure. The main theoretical result only applies to a very narrow case (existence of exact isometry), and the empirical gains are modest. The authors suggest applications to word embeddings, but these are not explored deeply.

---

> ### Author Rebuttal · Authors · 2025-07-31
>
> We appreciate your detailed and thoughtful review.
>
> - The paper has a clear theoretical goal. In particular, it defines a clean and interpretable optimization problem: select D columns from a P-column matrix to approximate an orthonormal basis.- The relaxation via multitask basis pursuit is standard but applied here in a targeted way. The main theorem (Proposition 4) states that if an orthonormal submatrix exists, it is in the support of the MBP solution. This is a good result under the assumption that such a submatrix exists.
> - The experimental comparisons with greedy show modest gains.
>
> Thank you!
>
> - Limited novelty: The method largely builds on known tools. In particular, the formulation is a direct application of group Lasso/multitask BP to a normalized design matrix. Also, the main theoretical guarantee is similar in spirit to known results in sparse recovery, especially in group sparsity contexts. The idea of trimming using convex relaxation before brute-force or greedy is standard in Lasso literature and feature selection. The paper feels more of a repackaging of known techniques and not a fundamentally new algorithmic or theoretical insight.
>
> The convex approach exhibited here is a fundamentally new algorithmic insight for isometric coordinate selection. Multitask basis pursuit, its application to matrix inversion, our ground truth loss, and our normalization method are not found elsewhere in the literature. Of course, we welcome specific suggestions of citations of relevant prior work. We agree that trimming is standard (Meinshausen https://www.sciencedirect.com/science/article/abs/pii/S0167947306004956, Hesterberg https://arxiv.org/abs/0802.0964, Koelle https://www.jmlr.org/papers/volume23/19-644/19-644.pdf).  However, we are only aware of Tibshirani (https://arxiv.org/abs/1206.0313) as presenting a rigorous argument for trimming (based on LARS selecting an l2 minimizer), and we are not aware of any result comparable to this for ADMM.
>
> - I found the motivation to be underdeveloped. The ties of the method proposed to interpretability, diversification, and embedding tasks are loose and not rigorously explored. For example, why is orthonormality the right or sufficient condition for interpretability or coordinate selection? How does this help in practice for downstream tasks (e.g., learning, generalization)? The introduction and discussion handwave at these points but do not really deliver on them, which weakens the practical relevance of the contribution.
>
> The desire for isometric embeddings is well-established (e.g. Isomap https://wearables.cc.gatech.edu/paper_of_week/isomap.pdf, Riemannian relaxation https://proceedings.neurips.cc/paper_files/paper/2016/file/cf1f78fe923afe05f7597da2be7a3da8-Paper.pdf, LDLE https://www.jmlr.org/papers/v22/21-0131.html) due to a desire for distances and angles to be preserved within embedding maps.  We will introduce additional discussion of relevant background on coordinate selection.  We agree that the link to diversification is not as well-established as we would like.  Our intent in including the diversification examples were to justify the use of Iris and Wine datasets and illustrate another application of the optimization problem, thereby extending the practical relevance past the embedding setting.
>
> - The theory only applies when a true isometry exists. but no bounds are given on how well the method performs when no exact orthonormal subset exists. This is a major limitation. For instance, is the solution approximately orthogonal? Is there a bound on the deviation from orthogonality? Without such results, the utility of the algorithm in the general case is unclear.
>
> An insightful bound on deviation from orthogonality, while desirable, is challenging to identify. The empirical success of Isometry Pursuit when the isometry condition is violated is the main support we present for the method in this case.
>
> - The experiments are too simplistic. The datasets (Wine, Iris, Ethanol) are small and toy-like. There is no compelling large-scale use case shown. Also, comparisons are limited to simple greedy, but no comparisons are made to more modern subset selection methods. The runtime analysis is limited and does not rigorously show why the proposed method is superior.
>
> The Ethanol (taken from Chmiela https://pubmed.ncbi.nlm.nih.gov/28508076/) originally accompanied a Nature article on computational chemistry, and is not a toy data set.  The dimensionalities explored are similar to those investigated in LDLE (https://www.jmlr.org/papers/volume22/21-0131/21-0131.pdf). Among subset selection methods for coordinate selection like LDLE and Chen (https://papers.nips.cc/paper_files/paper/2019/hash/6a10bbd480e4c5573d8f3af73ae0454b-Abstract.html), there is diversity amongst actual objective functions, so we feel that comparing against the general greedy paradigm is most informative.  We are open to comparing with any specifically suggested method.
>
> - There are several points where definitions or symbols are introduced abruptly or used vaguely. For example, the differential definitions (Def. 1 and 2) are not tightly integrated into the rest of the math. The normalization function q is defined in a confusing way.
>
> We will link definitions 1 and 2 more tightly with increased discussion of interpretable coordinate selection.  We will remove the parameter c from the normalization to simplify the function q.
>
> - Motivation: Why is orthonormality the right or sufficient condition for interpretability or coordinate selection? How does this help in practice for downstream tasks (e.g., learning, generalization)?
>
> See response above - we will introduce isometry more rigorously and review the literature in isometric coordinate selection.
>
> - Theory: Is the solution approximately orthogonal in the approximate case where no isometry exists? Is there a bound on the deviation from unitarity? Can you derive such results?
>
> See above response - unfortunately we can't promise this result right now, but will emphasize that our empirical results show that approximate orthogonality is tolerated well in practice.
>
> - Experiments: Comparisons to other subset selection methods
>
> See above response: numerous subset selection methods exist across literature.  To simplify, we have broken these down into greedy, brute, and convex.  We are open to incorporation of any specifically suggested method, but believe the usefulness of the method is demonstrated through comparison across these categories.

---

> > ### Comment · Reviewer_h74u · 2025-08-01
> > **Response to authors**
> >
> > Thank you for your responses. My concerns remain, especially regarding the limited novelty, the lacking motivation and the major limitation concerning orthogonality. Given these issues, I will stick to my rating.

---

### Official Review · Reviewer_mdRJ · 2025-07-03

**Clarity:** 1
**Significance:** 2
**Originality:** 2
**Rating:** 2
**Confidence:** 5

**Summary:**

The authors study the problem of column selection from a large-scale matrix with the goal of having the select submatrix orthogonal (i.e., an isometry). Using a formulation via a particular norm minimization in Equation 23, they propose their algorithm in page 6, called isometry pursuit. They prove in Proposition 3 that it is rotationally invariant, and  the optimal solution is always feasible (Proposition 4). The paper is concluded with experiments.

**Questions:**

This is an interesting paper. Unfortunately, I found the paper less polished and needs major changed to be prepared for publication. The novelty of the results are not well explained, and more evidence should be given for applications of the method, too.

### Major Comments/Question:


- Section 3.1 is not clear. Before it, everything is clear and easy to follow, but in this section, it is not clear why you need a ground truth objective. The role of parameter c is not well discussed, and why you pick such exponential (cosh?) function. Why the loss function is maximized for orthogonal matrices? Why it is invariant under inversion? There is also a typo in notations there: Please either use $l$ or $\ell$ (later is preferred in academic writing but it is important to keep consistency in the paper)

- What does Definition 3 want to say, in words? I know that you provided explanations there but I still have trouble to parse it. That part is difficult to follow, though it is essential for understanding the rest of the paper. Also, after Equation 8 you mentioned that q is convex, but this violated Definition 3.

- Algorithms are not labeled.

- It is a bit hard to understand what the contribution of the paper really is. The paper is trying to present the results via a story (which is great) but unfortunately the challenges and the original ideas used to resolve the problem are not well discussed and explained. The theory is more or less not deep enough, and the contributions look incremental (based on the current writing of the paper--otherwise I believe the paper has non-trivial interesting contributions that are probably not well phrased).

### Other Comments:


- In Equation 3, please remind the reader the definition of loss. Also, it would be great if you can have a couple of examples about different loss functions.


- Suggestion: for equation 3, use $\operatorname{argmin}$; this makes things look nicer

- After Equation 9, you say: "Regardless ... non-convex." What do you mean there, because the problem is combinatorial and convexity is not even well-defined there. Please clarify.

 - Equation 23: What does w mean? Please include explanation in the paper

**Ethical Concerns:**

["NO or VERY MINOR ethics concerns only"]

**Final Justification:**

The authors barely provide compelling responses to my concerns, and the rebuttal is mostly trying to accept the missing point and promising changes to the next version, thus I'm not convinces to raise my score. The concerns are still valid.

**Limitations:**

yes

**Paper Formatting Concerns:**

There is just a minor formatting issue as the paper does not contain line numbers. But I emphasize that this is minor and by no means do I want this comment to affect the paper.

**Quality:**

2

**Strengths And Weaknesses:**

Pros:

- motivating problem setting


Cons:

- paper's writing can be substantially improved; some equations can be made look nicer, more aligned; the paper looks a bit unpolished

- there is a minor issue with the paper formatting (and by no means I'm asking for rejecting for this--so this is just a heads up for the future modifcation). Line numbers are missing.

---

> ### Author Rebuttal · Authors · 2025-07-31
>
> Thank you for your detailed comments
>
> - paper's writing can be substantially improved; some equations can be made look nicer, more aligned; the paper looks a bit unpolished
> - there is a minor issue with the paper formatting (and by no means I'm asking for rejecting for this--so this is just a heads up for the future modifcation). Line numbers are missing.
>
> We will address these formatting issues in the revised version.
>
> - This is an interesting paper.
>
> Thank you!
>
> - Section 3.1 is not clear. Before it, everything is clear and easy to follow, but in this section, it is not clear why you need a ground truth objective. The role of parameter c is not well discussed, and why you pick such exponential (cosh?) function. Why the loss function is maximized for orthogonal matrices? Why it is invariant under inversion? There is also a typo in notations there: Please either use l or \ell (later is preferred in academic writing but it is important to keep consistency in the paper)
>
> We appreciate your concerns about Section 3.1 seeming somewhat arbitrary.  We divide these into three.  First, as mentioned in the paper (page 3), we introduce our new objective because nuclear norm and deformation (essentially, operator norm) are deficient for measuring isometry. Second, we introduce c for some control over the norms scaling, but given that this is not used in the experiments besides the rather uninteresting Figure 3 we will remove it in the revised version. Third, we will add propositions showing the questioned claims, and go into more detail about the LogSumExp substitution used to derive the loss function (which is not cosh - as far as we know it has no recognized name).  Thank you for catching the l versus \ell typo.  We will use \ell.
>
> - What does Definition 3 want to say, in words? I know that you provided explanations there but I still have trouble to parse it. That part is difficult to follow, though it is essential for understanding the rest of the paper. Also, after Equation 8 you mentioned that q is convex, but this violated Definition 3.
>
> As mentioned in the paper, Definition 3 introduces a criteria for normalization that is logarithmically symmetric around 1. Sorry, we should have said q is concave. Good catch.
>
> - It is a bit hard to understand what the contribution of the paper really is. The paper is trying to present the results via a story (which is great) but unfortunately the challenges and the original ideas used to resolve the problem are not well discussed and explained.  The theory is more or less not deep enough, and the contributions look incremental (based on the current writing of the paper--otherwise I believe the paper has non-trivial interesting contributions that are probably not well phrased).
>
> Multitask basis pursuit, its application to matrix inversion, our ground truth loss, and our normalization method are each novel, as is their application to coordinate selection and diversification.  We will make these contributions more explicit in the revised version.
>
> - In Equation 3, please remind the reader the definition of loss. Also, it would be great if you can have a couple of examples about different loss functions.
>
> Will do - is there an example you have in mind?
>
> - Suggestion: for equation 3, use \ell; this makes things look nicer
>
> Will do
>
> - After Equation 9, you say: "Regardless ... non-convex." What do you mean there, because the problem is combinatorial and convexity is not even well-defined there. Please clarify.
>
> Your concern is precisely what we were trying to explain.  We will clarify this.
>
> - Equation 23: What does w mean? Please include explanation in the paper
>
> w is defined in equation 15 - it is just the broadcasting of our normalization across the columns of a matrix.  We will provide a verbal explanation in the paper.

---

> > ### Comment · Reviewer_mdRJ · 2025-08-04
> > **Response**
> >
> > Dear Authors,
> >
> > I sincerely thank you for your response to my comments. I read them all in detail. I want to clarify that my question about the paper's contributions primarily concerned its writing. So I believe the results are interesting, but the writing makes them look incremental (see my original comment that you quoted). I strongly encourage you to edit the paper's writing, as these results are promising, and it would be great if they were also well-phrased. Also, thanks a lot for your answer to my other questions--please apply the updates to the next version of the paper.
> >
> > I decided to keep my score unchanged, as the changes that I feel are necessary are major. Please don't take my evaluations suggesting the end of this project, as I believe it will get accepted either in this venue or later on in other venues. Thanks!

---

### Official Review · Reviewer_K6aH · 2025-07-16

**Clarity:** 3
**Significance:** 2
**Originality:** 3
**Rating:** 3
**Confidence:** 2

**Summary:**

This paper proposes a new convex method called Isometry Pursuit for selecting approximately orthonormal submatrices from a wide matrix. The idea is to first normalize the column vectors in a specific way and then apply multitask basis pursuit to identify candidate subsets. The method is evaluated on synthetic and real datasets like Iris, Wine, Ethanol, and a small word embedding dataset. The authors provide both theoretical justification and empirical evidence that their method works well, especially when combined with a two-stage approach.

**Questions:**

1. Are there any more realistic examples to better illustrate the necessity or significance of the targeted problem? The example, “Say you are hosting an elegant dinner party, and wish to select a balanced set of wines for drinking and flowers for decoration,” may be too brief or abstract to fully convey the scenario.

2. Isometry Pursuit is a convex algorithm. Why can it produce non-unique solutions?

3. Could you analyze the time complexity of the proposed method compared with that of the greedy algorithm? It would also be helpful to provide some empirical runtime results.

4. Could you discuss the scalability of the proposed method, and how it could be applied in recommendation systems or RAG?

**Ethical Concerns:**

["NO or VERY MINOR ethics concerns only"]

**Final Justification:**

**Unresolved Issues:**

- The computational complexity of the proposed method remains unclear. While the authors argue that iteration counts are unknown a priori, a theoretical or empirical analysis (e.g., convergence rate or runtime experiments) would strengthen the paper.

- The motivation and real-world applicability could still be improved, as noted by Reviewer h74u. The current examples, though useful, lack depth in connecting to large-scale scenarios like recommendation systems.

**​Resolved Issues:**
​
- The non-uniqueness of Isometry Pursuit solutions is now clarified.

**Limitations:**

The primary limitation appears to be that the proposed method may not always find the optimal solution.

**Paper Formatting Concerns:**

None.

**Quality:**

3

**Strengths And Weaknesses:**

**Strengths:**

1. The method is backed by solid theoretical guarantees. In particular, Proposition 4 shows that the convex relaxation will retain any exactly orthonormal subset, if it exists.

2. The experiments show that the proposed two-stage method can outperform greedy baselines in terms of a ground truth isometry loss.

---

**Weaknesses:**

1. The time complexity of the two-stage method is not clearly stated, whereas the complexity of the brute-force algorithm is given as $O(C_l P^D)$.

2. Apart from efficiency, the scalability of the proposed algorithms is not discussed. In Section 5, one potential application mentioned is in recommendation systems, which typically involve large-scale data where scalability is critical.

3. It would be helpful for new readers if the paper included more examples to illustrate the importance of the target problem, i.e., selecting orthonormal column submatrices from wide matrices.

4. In Section 5, the paper refers to potential applications in recommendation systems and RAG. However, it is not clear how the proposed method could be concretely applied in such scenarios.

---

> ### Author Rebuttal · Authors · 2025-07-31
>
> Thank you for the thoughtful review
>
> - Are there any more realistic examples to better illustrate the necessity or significance of the targeted problem? The example, “Say you are hosting an elegant dinner party, and wish to select a balanced set of wines for drinking and flowers for decoration,” may be too brief or abstract to fully convey the scenario.
>
> We believe the four examples in the paper convey the applicability of the method.  Of course, we are receptive if you have a particular dataset in mind.
>
> - Isometry Pursuit is a convex algorithm. Why can it produce non-unique solutions?
>
> The objective is convex but not strictly convex, and so may produce non-unique solutions.
>
> - Could you analyze the time complexity of the proposed method compared with that of the greedy algorithm? It would also be helpful to provide some empirical runtime results.
>
> We cannot provide the computation complexity of the convex method since we are not aware a priori how many iterations will be required to fit the model. Empirical runtime results are provided in the supplement.
>
> - Could you discuss the scalability of the proposed method, and how it could be applied in recommendation systems or RAG?
>
> In RAG, diversification via greedy approaches like Maximum Marginal Relevance is somewhat standard, and basis pursuit may be part of a convex alternative. However, Isometry Pursuit would not apply without some non-trivial adjustments, including to enable selection of |S| < D coordinates to handle the increased scale.

---

> > ### Comment · Reviewer_K6aH · 2025-08-09
> >
> > Thank you for your response. However, some concerns remain unresolved, particularly regarding computational complexity. While the number of iterations isn't known a priori, it could be treated as a hyperparameter in complexity analysis or studied through convergence rate analysis and empirical validation.
> >
> > I also agree with Reviewer h74u that the paper's motivation could be strengthened further. The current examples, while helpful, could benefit from more concrete real-world applications to better demonstrate the problem's significance.

---

### Note · Authors · 2025-08-16

We thank the reviewers for their kindhearted and useful feedback and will continue to work on the manuscript.

---

### Decision · Program_Chairs · 2025-09-17

**Decision:**

Reject

**Comment:**

Post rebuttal, the reviewers still have strong concerns for the paper. I agree computational complexity and real-world applicability are two major concerns still. I must reject the paper.